# PPyNT/NR/NBR Composites with Excellent Microwave Absorbing Performance in X-Band

**DOI:** 10.3390/polym15081866

**Published:** 2023-04-13

**Authors:** Huiru Yang, Aiping Wang, Xincong Feng, Hailing Dong, Tao Zhuang, Jing Sui, Shugao Zhao, Chong Sun

**Affiliations:** 1Key Laboratory of Rubber-Plastics, Ministry of Education/Shandong Provincial Key Laboratory of Rubber-Plastics, Qingdao University of Science & Technology, Qingdao 266042, China; yhr1150286872@163.com (H.Y.);; 2School of Materials Science and Engineering, Ocean University of China, Qingdao 266100, China

**Keywords:** polypyrrole nanotube, immiscible rubber blend, impedance matching, microwave absorption

## Abstract

To meet the comprehensive demand for flexible microwave absorbing (MA) materials, a novel MA rubber containing homemade Polypyrrole nanotube (PPyNT) is produced based on the natural rubber (NR) and acrylonitrile-butadiene rubber (NBR) blends. To achieve the optimal MA performance in the X band, the PPyNT content and NR/NBR blend ratio are adjusted in detail. The 6 phr PPyNT filled NR/NBR (90/10) composite has the superior MA performance with the minimum reflection loss value of −56.67 dB and the corresponding effective bandwidth of 3.7 GHz at a thickness of 2.9 mm, which has the merits in virtue of achieving strong absorption and wide effective absorption band with low filler content and thickness compared to most reported microwave absorbing rubber materials over the same frequency. This work provides new insight into the development of flexible microwave-absorbing materials.

## 1. Introduction

With X-band (8.2–12.4 GHz) frequency electromagnetic waves being extensively utilized in wireless networks, satellite communications, and radar technology, adverse effects caused by electromagnetic radiation on human health and communication security are becoming a severe problem worldwide [1]. Furthermore, the successive advent of radar-detecting technology has greatly reduced the survivability of military equipment in modern warfare [2]. Therefore, materials capable of absorbing microwaves over the X-band are highly demanded in both civil and military fields.

Typical characteristics of ideal microwave absorbing (MA) materials include lightweight, strong absorption, and wide effective absorption band (EAB). Among various reported MA materials, rubber-based microwave absorbers represent one class of most promising materials in practical application due to their high flexibility, excellent environmental resistance, desirable mechanical properties, good versatility, and ease to process [3]. Generally, rubber-based microwave absorbers are fabricated by incorporating absorbing fillers into rubber matrices. Depending on actual application circumstances, rubber matrices vary in natural rubber (NR) [4], butadiene rubber (NBR) [5], hydroacrylic butadiene rubber (HNBR) [6], silicon rubber (SR) [7], etc. Magnetic materials such as magnetic alloys [2] and ferrite [5], among other materials, are popular absorbing fillers due to their high capability to attenuate electromagnetic waves via magnetic loss. However, the corresponding rubber absorbers normally suffer from high weight and poor mechanical properties due to the high filler density and required filler content. Carbon nanomaterials (such as carbon black [8], carbon nanotubes [9], graphene [7], reduced graphene oxide [4], etc.) also have been extensively exploited in fabricating rubber absorbers because of low density and high dielectric loss capacity. Although carbon-based nano-fillers can endow rubber matrices absorption capability to a certain degree with relatively lower filler loading, the related rubber absorbers normally face the problem of narrow EAB resulting from poor impedance matching [10]. Hybrids of magnetic materials and carbon nanomaterials are becoming the recent interest because of their dual loss mechanism and better impedance matching. Numbers of rubber-based microwave absorbers, e.g., Fe-Ni/C/NBR [11], Fe_3_O_4_/rGO/SR [3], Sr-ferrite/NBR [5], etc., have been successfully developed for application in the X-band frequency range. However, most of these products involve complicated fabrication procedures unsuitable for large-scale production. In addition, a relatively high filler loading (over 40 wt.%) [3] is required to attain sufficient absorption performance.

Apart from filler properties, filler dispersion is another important factor affecting rubber absorbers’ MA performance. Well-established filler networks inside the rubber matrix are favorable for excellent performance. Earlier efforts have mainly focused on the uniform dispersion of fillers in a single rubber matrix. Recently, an immiscible rubber blend has emerged as a promising alternative in the design of rubber-based microwave absorbers because of the enhanced filling efficiency resulting from the selective distribution of fillers in a specific phase [12]. In addition, the heterogeneous rubber blend composites possess more effective multiple internal scattering, including inter-filler and interphase scattering, than single rubber composite, which only has inter-filler scattering [13]. A MA rubber based on a partially immiscible blend of NR and epoxidized NR containing 30 wt.% conductive carbon blacks has been successfully developed by Li.et.al [14], which shows much stronger absorption capability compared to either of single rubber-based composites containing the same amount of filler. However, this MA rubber composite is non-effective over the 8.2–12.4 GHz frequency range.

Despite lots of efforts devoted to the development of MA rubber, there is still a huge blank in designing a new material system to achieve strong absorption capability and a wide effective absorption band over the frequency range of 8.2–12.4 GHz with low filler content. Polypyrrole (PPy), one of the typical intrinsically conducting polymers (ICPs), has been investigated as Dielectric loss-type MA material extensively from different aspects, including preparation method [15], morphology [16], hybrid with other materials [17], etc. However, most studies mainly focus on the MA performance of separate powders in paraffin, far from practical applications. Herein, we report a novel MA rubber by incorporating homemade PPy nanotube (PPyNT) into the immiscible blend of natural rubber (NR) and acrylonitrile-butadiene (NBR) through a simple mixing technology in the rubber industry based on our previous work [12]. By simply adjusting the PPyNT content and rubber blend ratio, the impedance matching and attenuation capability of the obtained MA rubber are tuned. The synergistic effect of PPyNT itself, its unique distribution behavior in immiscible NR/NBR blend, and multiple interfaces structure in the rubber blend is made good use of, which leads to a strong reflection loss value of −56.67 dB and broad effective bandwidth of 3.7 GHz at 2.9 mm in X band when the PPyNT content is 6 phr, and the NR/NBR blending ratio is 90/10. This result surpasses most rubber-based absorbers reported in lots of literature over the same frequency range. In addition, a possible microwave absorbing mechanism for the optimal composite in this study is proposed. This work provides new insight into the design of rubber-based absorbers in the X band.

## 2. Materials and Methods

### 2.1. Materials

Pyrrole, ferric chloride hexahydrate (FeCl_3_·6H_2_O), and methyl orange (MO) were purchased from Aladdin Biochemical Technology Co., Ltd. (Shanghai, China). Natural rubber and acrylonitrile-butadiene rubber with 33 wt.% acrylonitrile content were obtained from Nandi Chemical Industry Co., Ltd. (Chongqing, China). Stearic acid (SA), zinc oxide (ZnO), sulfur (S), poly(1,2-dihydro-2,2,4-trimethyl-quinoline) (RD), and N-tert-butylbenzothiazole-2-sulphenamide (NS) were provided by Guan Lian polymeric Material Co., Ltd. (Taicang, China). Pyrrole was distilled under pressure before use, and other materials were used without purification.

### 2.2. Synthesis of PPyNT

PPyNT was synthesized through a reactive self-degraded template method, according to a previous report [18]. Firstly, 0.03 mol pyrrole and 0.0015 mol MO were dissolved in 100 mL of distilled water. Then, 100 mL of 0.06 mol FeCl_3_ aqueous solution was dropped into the pyrrole solution. The mixing solution was magnetically stirred for 12 h at room temperature. The product was filtered, washed with ethanol and distilled water several times, and dried in a vacuum oven at 60 °C for 24 h.

### 2.3. Fabrication of PPyNT/NR/NBR Composites

First, 100 phr (parts per hundred weights concerning rubber) rubber blend with different NR/NBR weight ratios from 10/90–90/10 was mixed using a two-roll open mill (BL-6175-AL, Baolun, China) for 60 s with a roll spanning of 1 mm and a friction ratio of 25/18. Then 3 phr ZnO, 1 phr SA and 1 phr RD were added into the mill simultaneously and blended for 300 s. After that, different weight of PPyNT (3 phr–12 phr) was introduced into the mill and blended for 120 s. Then 1.5 phr NS and 2 phr S were added into the chamber subsequently and blended for 120 s. The temperature was maintained at around 40 °C during the entire blending process. Following the blending process, the compounds were vulcanized under pressure at 160 °C for the optimum vulcanization time (t_90_), obtained from a moving die rheometer (RHEOMETER MDR 2000, ALPHA Technologies, Wilmington, DE, USA).

### 2.4. Characterization

The morphology of PPyNT was investigated by Field Emission Scanning Electron Microscope (FESEM, JSM-7500F, JEOL, Tokyo, Japan) and Transmission Electron Microscope (TEM, JEM-2100, JEOL, Tokyo, Japan). Structural information of PPyNT was obtained with X-ray Diffractometer (XRD, D/Max 2500V/PC, Rigaku Corporation, Tokyo, Japan) and Fourier-transform infrared (FTIR, VERTEX70, Bruker, Bremen, Germany) spectrometer. The conductivity of PPyNT was measured by a four-probe conductivity meter (ST2253, Jingge Electronic, SuZhou, China). To analyze the phase morphology and filler dispersion of PPyNT/NR/NBR composites, the ultra-thin sections of composites were cut by ultramicrotome (LEICAEM FC7, Leica Microsystems, Wetzlar, Germany) at −100 °C and then characterized by TEM. The electromagnetic parameters of rectangular PPyNT/NR/NBR composites with dimensions of 22.86 mm × 10.80 mm × 2 mm were measured by waveguide method at room temperature on ZNB-20 vector network analyzer (Rhodes & Schwarz, Munich, Germany) in X-Band frequency range (8.2–12.4 GHz).

## 3. Results and Discussion

### 3.1. Morphology and Structure of PPyNT

PPyNT is oxidatively polymerized through the reactive self-degraded template method, as shown in Figure 1a. With the addition of FeCl_3_ into the solution of MO and pyrrole, fibrillar MO-Fe^3+^ complexes are formed in the reaction system [19]. These MO-Fe^3+^ complexes induce the oxidation of pyrrole monomer on their surface and then degrade automatically with polymerization proceeding due to the reduction of Fe^3+^ by pyrrole monomer [19]. Eventually, the obtained polymer assembles into nanotubes. The morphological characteristics of as-synthesized PPyNT are investigated by SEM and TEM. Uniform tubular nanostructures with very high yield (Figure 1b) and rough surfaces (Figure 1c) are observed in SEM images owing to the adhesion of some granules. These tubular nanostructures are several micrometers in length with an average diameter of 104.2 ± 18.4 nm (Appendix A). The hollow feature of the tubular nanostructures is confirmed by TEM images (Figure 1d,e), where the average thickness of the wall is statistically determined as around 31.4 ± 3.4 nm (Appendix A). XRD and IR investigate the structure of as-synthesized PPyNT. A broad peak assigned to the π–π interaction of the polypyrrole chain is observed at 24.1° in the XRD pattern (Figure 1f) [20] and numbers of typical bands of PPy [21], including the ring skeletal vibration of PPy at 1533 cm^−1^ and 1453 cm^−1^, the =C–H in-plane vibration at 1292 cm^−1^, the C–N stretching vibration at 1160 cm^−1^, the C–H/N–H in-plane vibration at 1027 cm^−1^, the C–C out-plane deformation vibration at 962 cm^−1^ and the C–H wagging at 768 cm^−1^, are observed in IR spectrum (Figure 1g). The conductivity of PPyNT measured by a four-probe meter is about 0.35 S/cm. The hollow structure can enrich multi reflection to facilitate microwave attenuation, and large aspect ratios can promote fair filler connection in rubber matrix [22]. All these features imply that the obtained PPyNT may serve as a suitable absorbing filler to develop Rubber-Based microwave absorbers.

### 3.2. MA Performance of PPyNT/NR/NBR Composites

Normally, an excellent absorber should allow the incident electromagnetic wave to penetrate it to the largest extent and simultaneously attenuate the entering wave within the limited thickness [23]. In the case of rubber composites, absorbing fillers modify the electromagnetic parameters of composites to fulfill these requirements. Therefore, their content in the rubber matrix plays a vital role in deciding the final MA performance. Meanwhile, different phase morphologies such as co-continuous and matrix-droplet structures can be formed by varying the blend ratio of rubber matrices, which greatly impact the final properties of the composites [24]. In this context, two decisive factors (i) PPyNT content and (ii) the blend ratio of NR/NBR are investigated in detail to gain the optimal PPyNT/NR/NBR composite with excellent MA performance.

#### 3.2.1. Optimization of PPyNT Content

Different contents of PPyNT ranging from 3 to 12 phr are impregnated into NR/NBR (50/50). With this blend ratio, composites show a typical co-continuous phase structure, as shown in Appendix A, where the bright and dark regions are NR and NBR phases, respectively [25]. Meanwhile, PPyNTs are selectively localized in the NR phase regardless of their content.

The MA performance of an absorber highly relies on its complex permittivity ($\varepsilon_{r}=\varepsilon^{\prime }-j\varepsilon^{\prime \prime }$) and permeability ($\mu_{r}=\mu^{\prime }-j\mu^{\prime \prime }$). ε′ and μ′ correlate with the capacity of the material in storing electrical and magnetic energy, respectively, while ε″ and μ″ are related to its capability to dissipate electrical and magnetic energy, respectively [17]. Because of their non-magnetic nature, the μ′ and μ″ values of all samples are almost 1 and 0, respectively. Therefore, only the complex permittivity is analyzed in detail here. As seen in Appendix A, the ε′ and ε″ of all samples show a typical frequency dispersion behavior, descending with an increase in frequency [26]. With increased PPyNT content, this frequency dispersion behavior is becoming more remarkable. At the same time, the values of ε′ and ε″ increase clearly with the content of PPyNT rising, which confirms the effective medium theory [27]. On one hand, with the increase in PPyNT content, there are more mini-capacitors with PPyNT as the electrode and a very thin NR layer in between as dielectric [28] in the composites. On the other hand, a more interconnected PPyNT network tends to be formed in the composites with higher PPyNT content, as shown in Appendix A, which is able to offer more paths to facilitate the electron migrating or hopping and in turn induce leakage current [22]. Consequently, the enhanced capability in storing and dissipating electromagnetic energy is observed simultaneously with increased PPyNT content.

The reflection loss (*RL*) and effective absorption bandwidth (EAB) are employed to evaluate the MA performance of the composites. The values of *RL* are calculated by Appendix A based on the transmission line theory [12]. The criterion for the EAB is the value of *RL* less than −10 dB, meaning over 90% of energy is absorbed. The minimum *RL* (*RL*_min_) of the composite containing 3 phr PPyNT is only −9.8 dB at 11.2 GHz with a thickness of 3.4 mm (Figure 2a), indicating its weak absorbing capability. The composite with 6 phr PPyNT shows the best absorbing performance. Its *RL*_min_ value can reach −37.94 dB at 8.3 GHz with a thickness of 3.4 mm (Figure 2b), and the corresponding EAB is 1.6 GHz (from 8.2–9.7 GHz) (Figure 2(b′)). The thinner thickness of 2.8 mm can achieve the widest EAB of 3.4 GHz (from 8.8–12.1 GHz), covering 79% of the X-band (Figure 2(b′)). Further increase in PPyNT content does not favor the improvement in MA performance. The *RL*_min_ value of the composite with 9 phr PPyNT is −12.84 dB with a thickness of 2.2 mm (Figure 2c) and a corresponding EAB of 2.8 GHz (from 9.7–12.4 GHz) (Figure 2(c′)). The composite with 12 phr PPyNT doesn’t show effective absorption over the whole X-band, and its *RL*_min_ value is only −7.84 dB at 9.7 GHz (Figure 2d).

For a better understanding of the reason for the variation of absorbing performance along with PPyNT content, the dielectric loss tangent (tanδε = ε″/ε′) and the attenuation constant α (described by Appendix A) of different composites are compared. Tanδε refers to the ability of materials to dissipate electromagnetic energy via dielectric loss, while the attenuation constant reflects the comprehensive attenuating capability of materials [17]. As seen clearly in Figure 3a,b, both tanδε and α rise gradually with the increase in PPyNT content, suggesting the enhanced attenuating capability for the composite with higher PPyNT content. It is contrary to the observation in their MA performance, which is depressed while PPyNT content is over 6 phr. This result is attributed to the poor impedance matching ($\left|Z_{\mathrm{in}}/Z_{0}\right|$) resulting from excessive fillers. Impedance matching reflects the extent of electromagnetic waves entering the interior of the material and is calculated according to Appendix A [29]. The value of $\left|Z_{\mathrm{in}}/Z_{0}\right|$ closer to 1, the better impedance matching [17]. Figure 3c–f shows contour maps of $\left|Z_{\mathrm{in}}/Z_{0}\right|$ values for PPyNT/NR/NBR (50/50) with different PPyNT contents. It is clear that the composite with 6 phr PPyNT has the largest area with $\left|Z_{\mathrm{in}}/Z_{0}\right|$ values approaching 1 (Figure 3d) among all samples. Further increase in PPyNT content results in a greater deviation from 1 in $\left|Z_{\mathrm{in}}/Z_{0}\right|$. Overall, owning to the good balance achieved between attenuation capability and impedance matching, the PPyNT/NR/NBR (50/50) composite with 6 phr PPyNT is near optimal for achieving an excellent MA performance.

#### 3.2.2. Optimization of the NR/NBR Blend Ratio

Based on the optimal PPyNT content obtained above, the blend composition is further optimized by changing the NR/NBR ratio from 10/90 to 90/10. Obviously, the phase morphologies of 6 phr PPyNT filled NR/NBR composites are great related to the ratio of NR/NBR. A matrix-droplet structure with NR phase as droplet and NBR as matrix is found in the samples with NR/NBR ratio of 10/90 (Appendix A) and 30/70 (Appendix A). The size of the NR phase increases with increasing the content of NR, and a typical co-continuous structure is formed in the composite while the ratio of NR/NBR reaches 50/50 (Appendix A). With further increasing the NR/NBR ratio to 70/30 (Appendix A) and 90/10 (Appendix A), the NR and NBR phases change to matrix and droplet, respectively, and the size of the droplet NBR phase decreases with increasing the NR content as well. The alteration in blend ratio has little effect on the selective localization of PPyNT in the NR phase.

The ε′, ε″ and tan*δ_ε_* for 6 phr PPyNT filled NR/NBR composites with different NR/NBR ratios are shown in Figure 4a–c. The values of ε′ range from 8.11–7.68, 8.38–7.66, 7.94–7.05, 5.95–5.12 and 7.24–6.26 for NR/NBR ratio of 10/90, 30/70, 50/50, 70/30 and 90/10 respectively. The values of ε″ for the corresponding composites range from 2.46–2.20, 2.54–2.59, 2.93–2.77, 3.27–2.75, and 3.06–2.85, respectively. Accordingly, the capability of dielectric loss (tan*δ_ε_*) for the composites follows the order (NR/NBR): 70/30 > 90/10 > 50/50 > 30/70 > 10/90. It is widely accepted that dielectric loss stems from conduction loss and polarization loss [23]. In the case of the dielectric loss primarily originating from the conduction loss, the values of ε″ will decline with increasing frequency [30]. While the polarization loss is present, fluctuation and resonance peaks in ε″ curves can be observed [31]. It is noted that the ε″ values of the composites with NR/NBR = 90/10, 70/30, and 50/50 show an overall descending trend with increasing frequency accompanied by multiple peaks centered around 8.8, 9.5, 10.3, 11.1, and 11.9 GHz. Moreover, the decline of ε″ with frequency is more obvious for the NR/NBR = 70/30 composite, and simultaneously resonance peaks in ε″ are less visible for this sample. As for the two composites with lower NR/NBR ratio, i.e., 30/70 and 10/90, the values of ε″ almost lose the frequency dispersion feature except for the increment caused by resonance peaks. In particular, a strong resonance peak is observed at 8.6 GHz for the NR/NBR = 10/90 composite. These results suggest that polarization loss is present for all composites. This polarization loss may come from dipolar polarization caused by intrinsic dipoles of PPyNT [32] and interface polarization caused by structural heterogeneity in the composites (abundant interface between PPyNT and NR matrix) [23]. The conduction and polarization losses contribute to dielectric loss differently with varying NR/NBR ratios. The Cole-Cole curves based on Debye theory (expressed by Appendix A) [9] are employed to better understand the individual contribution from this two-loss mechanism of different samples. It is reported that one semicircle in the Cole-Cole curve corresponds to the process of a Debye relaxation, and a straight line stands for the conduction loss [33]. The radius of the semicircle and the slope of the straight line are related to the intensity of the relaxation process and the capacity of conduction loss, respectively [34]. As shown in Figure 4d, the Cole-Cole curves of the composites with NR/NBR = 90/10, 70/30, and 50/50 are closer to a straight line overall despite the appearance of several small semicircles, suggesting the predominant role of conduction loss in the dielectric loss for these samples. Especially the steeper slope of the line for the NR/NBR 70/30 composite indicates its higher conduction loss. On the contrary, polarization loss dominates dielectric loss of the NR/NBR of 30/70 and 10/90 composites, which show an undulant line with a slope approximately 0 and an obvious semicircle with a large radius, respectively. As we discussed above, a PPyNT/NR droplet-NBR matrix phase structure was formed in the composites with lower NR content. The conduction network formed by PPyNT was interrupted, which in turn contributes negatively to the conduction loss [24].

Figure 5 shows the three-dimensional *RL* plots of the composites with different NR/NBR ratios. The *RL*_min_, EAB, and the corresponding thickness are also summarized and presented. The results reveal that composite with higher NR content has better MA performance. The *RL*_min_ of the composite with the NR/NBR ratio of 10/90 and 30/70 is −18.25 dB at 3.2 mm (Figure 5a) and −19.28 dB at 2.3 mm (Figure 5b), respectively, while *RL*_min_ of the composites with rich NR content are all below −30 dB (Figure 5c–e). In particular, the *RL*_min_ of the composite with an NR/NBR ratio of 90/10 can reach −56.67 dB at 2.9 mm (Figure 5e), and the corresponding EAB can achieve 3.7 GHz (covering 86% of the X-band). The synergy between attenuation capability and impedance matching accounts for the observed variation in MA performance with a blend ratio of NR/NBR. Appendix A shows the attenuation constant (α) of different composites, where the order of α is (NR/NBR) 70/30 > 90/10 > 50/50 > 30/70 > 10/90, is consistent with that of dielectric loss tangents (Figure 4c). The $\left|Z_{\mathrm{in}}/Z_{0}\right|$ mappings of the composites are presented in Appendix A. Apparently, poor impedance matching is observed for the composites with lower NR content. The increase in NR content improves impedance matching. In particular, the superior impedance matching is observed for the composite with NR/NBR ratio of 90/10, which shows the largest area much approaching 1 in $\left|Z_{\mathrm{in}}/Z_{0}\right|$ mapping. Therefore, combined with its attenuation capability, 6 phr PPyNT filled NR/NBR (90/10) composite shows the best comprehensive MA performance. Compared with other rubber-based microwave absorbers reported in the literature (Table 1), it has the merits of achieving strong absorption and a broad effective absorption band with low filler content and thickness.

Based on the above discussion, a possible microwave absorbing mechanism for the 6 phr PPyNT filled NR/NBR (90/10) composite is briefly outlined in Figure 6. Firstly, an excellent impedance matching achieved with optimal composite composition facilitates the entry of more electromagnetic waves into the composite. Secondly, the conductive pathways formed by PPyNT in the continuous NR phase are beneficial for electron migrating or hopping, inducing conduction loss. Thirdly, the dipolar polarization originated from PPyNT, and interfacial polarization induced from the abundant heterogeneous PPyNT/NR interface also makes some contributions to attenuate electromagnetic energy. Fourthly, the PPyNT and NBR droplets are distributed in the NR matrix, providing multi-scattering and reflection sites to prolong the propagation path of electromagnetic waves and, in turn, benefit from energy attenuation.

## 4. Conclusions

In this study, PPyNTs with several micrometers in length, 104.2 ± 18.4 nm in diameter, 31.4 ± 3.4 nm in wall thickness, and 0.35 S/cm in conductivity are prepared from a reactive self-degraded template method. Microwave absorbing rubber based on PPyNT as absorbing filler and immiscible NR/NBR blend as matrix is fabricated via a simple mixing technique in the rubber industry. To optimize the material system achieving excellent MA performance in the frequency of 8.2–12.4 GHz, the PPyNT content and the ratio of NR/NBR are investigated in detail. Results show that the selective distribution of PPyNT in the NR phase is regardless of PPyNT content and NR/NBR ratio. In contrast, the impedance matching and attenuation capability of the composites are regulated by composite composition. The 6 phr PPyNT filled NR/NBR (90/10) has the best comprehensive MA performance, which shows a strong reflection loss value of −56.67 dB and a broad effective bandwidth of 3.7 GHz at 2.9 mm, surpassing most rubber-based microwave absorbers reported in the literature over the same frequency range. Based on the electromagnetic loss mechanism analysis, its excellent absorption performance can be attributed to superior impedance matching, conduction loss, dipolar polarization, interface polarization, and multiple scattering. This novel microwave-absorbing rubber composite is a promising candidate for high-performance microwave absorption material in radar stealth and flexible electronics.

## Figures and Tables

**Figure 1 polymers-15-01866-f001:**
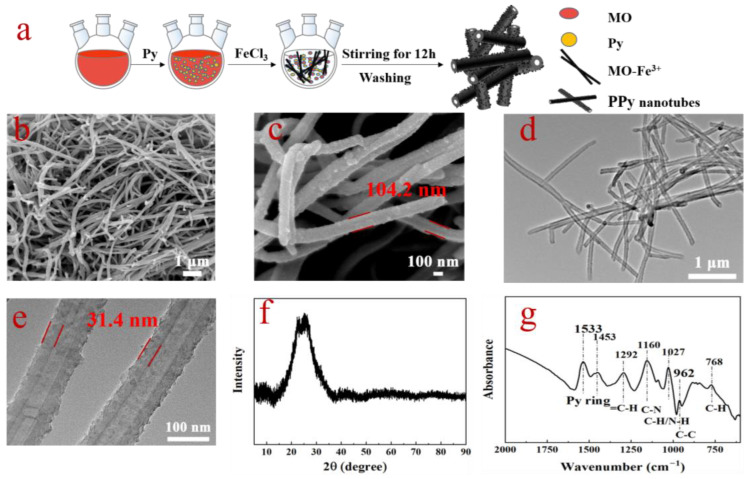
(**a**) The schematic illustration of synthesis of PPyNT; (**b**,**c**) SEM images, (**d**,**e**) TEM images, (**f**) XRD pattern and (**g**) IR spectrum of typical PPyNT.

**Figure 2 polymers-15-01866-f002:**
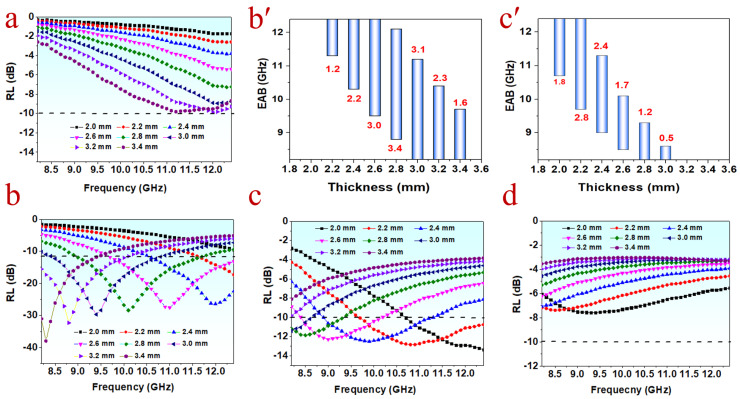
RL plots of PPyNT/NR/NBR (50/50) with (**a**) 3 phr PPyNT, (**b**) 6 phr PPyNT, (**c**) 9 phr PPyNT, (**d**) 12 phr PPyNT and EAB of PPyNT/NR/NBR (50/50) with (**b’**) 6 phr PPyNT and (**c’**) 9 phr PPyNT.

**Figure 3 polymers-15-01866-f003:**
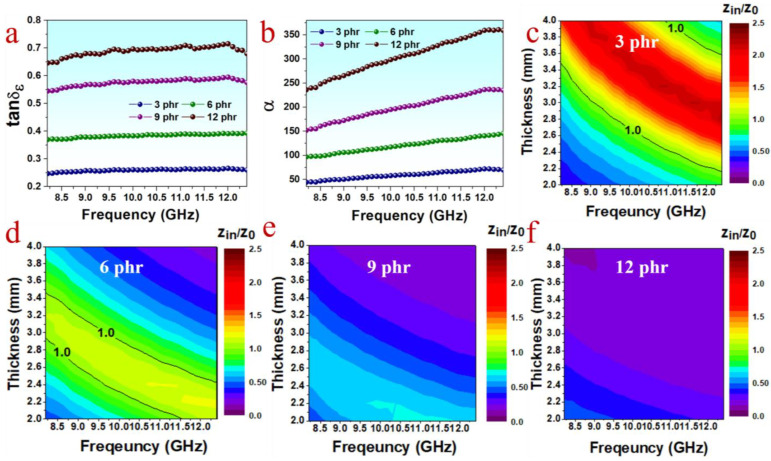
(**a**) Dielectric loss tangent factor, (**b**) attenuation constant and (**c**–**f**) 2D contour maps of $\left|Z_{\mathrm{in}}/Z_{0}\right|$ values for PPyNT/NBR/NBR (50/50) with different PPyNT contents.

**Figure 4 polymers-15-01866-f004:**
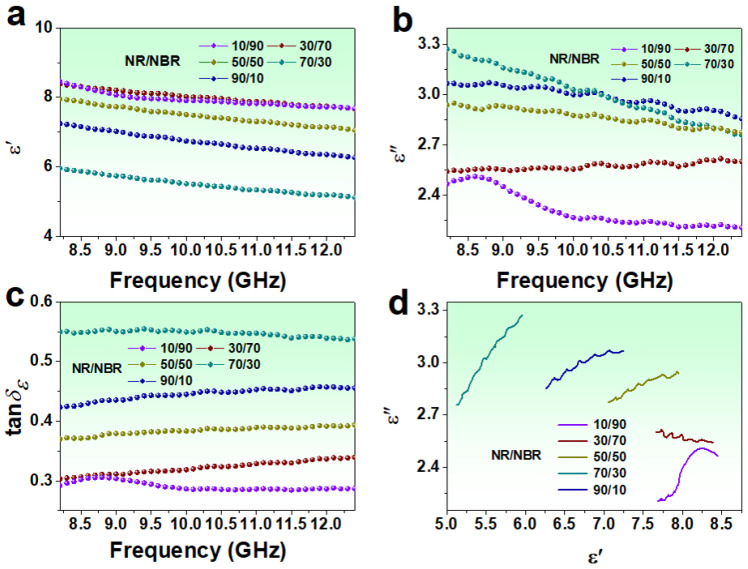
The real ε′ (**a**) and imaginary ε″ (**b**) parts of the complex permittivity, and the dielectric loss tangent factor tanδε (**c**); Cole-Cole curves (**d**) of 6 phr PPyNT filled NR/NBR composites with different ratios of NR/NBR.

**Figure 5 polymers-15-01866-f005:**
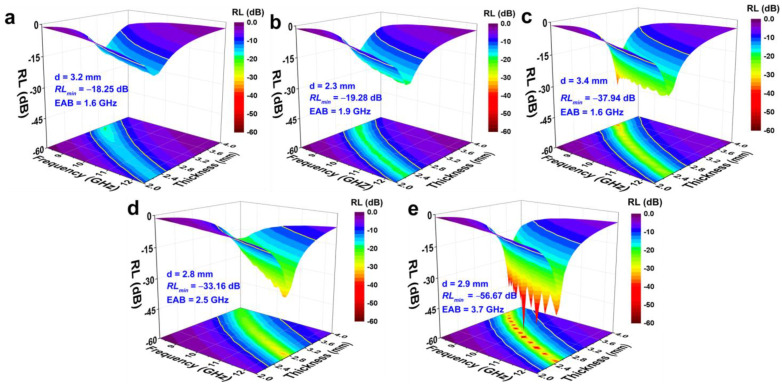
Three-dimensional RL plots for 6 phr PPyNT filled NR/NBR composites with different ratios of NR/NBR: (**a**) 10/90; (**b**) 30/70; (**c**) 50/50; (**d**) 70/30; (**e**) 90/10.

**Figure 6 polymers-15-01866-f006:**
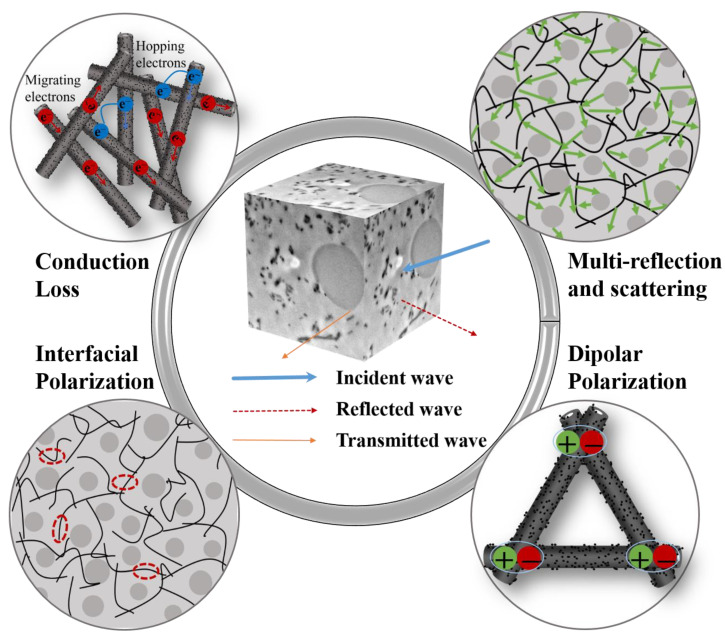
Schematic illustration of microwave absorbing mechanism for 6 phr PPyNT filled NR/NBR (90/10) composite.

**Table 1 polymers-15-01866-t001:** Comparison of the MA performance of the PPyNT filled NR/NBR (90/10) composite with other rubber-based absorbers in the frequency range of 8.2–12.4 GHz.

Sample	RL_min_ dB	EAB GHz	Thickness mm	Filler Loading (wt.%)	Ref.
Fe-Ni/C/NBR	−8~−10	0	1.5	70	[11]
CBSrF/NBR	−16	1.5	5	20.73	[5]
CB/NR	>−5	0	5	14.2	[8]
CCB/NR/ENR (70/30)	>−10	0	5.5	30	[14]
MWCNT/HNBR	−21.3	Not mentioned	2	8.26	[6]
rGO@Fe_3_O_4_/SR	−22.5	2.1	1.5	50	[3]
NiZn/TPNR	−38.3	Not mentioned	7	12	[35]
RGO/NR	−57	3.8	3	10	[4]
PPyNT/NR/NBR (90/10)	−56.67	3.7	2.9	5.24	This work

## Data Availability

Data presented in this study are available on request from the corresponding author.

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
