# Peer review of "PPyNT/NR/NBR Composites with Excellent Microwave Absorbing Performance in X-Band"

_polymers, 2023, doi:10.3390/polym15081866_

Round 1
Reviewer 1 Report
The present paper deals with preparing and investigating composites for microwave absorption properties and applications.
The paper is well prepared and written, however, small shortcomings remain. The paper can be accepted for publishing after minor revision:
1) The abbreviations must be discussed the first time, also in the abstract. For example, PPyNT, NT, PPy also others.
2) Lines 99-103 must be deleted.
3) Used equipment producer is missing for several methods. For example, rolls, mills, and others. Also, conditions must be given: time, temperatures of both rolls, mixing sequence - or batch, and other important details for repeatability of the experiments.
The overall discussion and figures quality are very good. The conclusions are adequate.
Author Response
Thank you very much for the helpful and constructive comments on our manuscript entitled “PPyNT/NR/NBR composites with excellent microwave absorb-ing performance in X-band”.
In this version, we have revised the manuscript according to your kind comments. Please check the list below for a detailed point-by-point response.
Comment 1: The abbreviations must be discussed the first time, also in the abstract. For example, PPyNT, NT, PPy also others.
Response 1: We have added the abbreviation of Polypyrrole nanotube (PPyNT) where PPyNT first appears in the journal as requested, namely in the abstract.
Comment 2: Lines 99-103 must be deleted.
Response 2: This section has been removed from the revised manuscript.
Comment 3: Used equipment producer is missing for several methods. For example, rolls, mills, and others. Also, conditions must be given: time, temperatures of both rolls, mixing sequence - or batch, and other important details for repeatability of the experiments.
Response 3: We have included the content in the revised manuscript (Part 2.3), including equipment producer of two-roll open mill and temperatures of both rolls, mixing time and sequence during the blending process.

Reviewer 2 Report
The manuscript entitled "PPyNT/NR/NBR composites with excellent microwave absorbing performance in X-band" investigated a novel microwave absorbing rubber based on the natural rubber (NR) and acrylonitrile-butadiene rubber (NBR) blends containing PPy nanotube. The paper presents an interesting discussion of the development of flexible microwave-absorbing materials. The authors have done a lot of good work. The explanation of the phenomena observed is quite satisfactory whereas the text is fairly well written and exhibits an element of originality. However, before publication in Polymers, some English verification is recommended, and incorporating the following suggested corrections would help the manuscript in meeting the journal publication standards.
- In Table 1, the contents of ingredients for all the samples are the same. Therefore, I highly recommend omitting the ingredient columns and simply adding the amount of each ingredient into the text (part 2.3).
- Please, put the unit on the axis of each graph into the parenthesis like in Figure 1g, and delete the slashes.
- In my view, the conclusion must be included more important results. Moreover, the authors should mention the potential application of this novel rubber-based microwave-absorbing material once more in the conclusions part.
Author Response
Thank you very much for the helpful and constructive comments on our manuscript entitled “PPyNT/NR/NBR composites with excellent microwave absorb-ing performance in X-band”.
In this version, we have revised the manuscript according to your kind comments. Please check the list below for a detailed point-by-point response.
Comment 1: In Table 1, the contents of ingredients for all the samples are the same. Therefore, I highly recommend omitting the ingredient columns and simply adding the amount of each ingredient into the text (part 2.3).
Response 1: We have added the content of each ingredient into the text in revised manuscript (Part 2.3) according to reviewer’s comment.
Comment 2: Please, put the unit on the axis of each graph into the parenthesis like in Figure 1g, and delete the slashes.
Response 2: We have put the units on the axes of all graphs into the parenthesis, including the revised manuscript and supporting information.
Comment 3: In my view, the conclusion must be included more important results. Moreover, the authors should mention the potential application of this novel rubber-based microwave-absorbing material once more in the conclusions part.
Response 3: We have added more major results in the conclusion and mentioned the potential application of the reported rubber composite as high performance microwave absorbing material in radar stealth and flexible electronics.
